# Morph-fitting: Fine-Tuning Word Vector Spaces with Simple Language-Specific Rules

## Abstract

Morphologically rich languages accentuate two properties of distributional vector space models: 1) the difficulty of inducing accurate representations for low-frequency word forms; and 2) insensitivity to distinct lexical relations that have similar distributional signatures. These effects are detrimental for language understanding systems, which may infer that *inexpensive* is a rephrasing for *expensive* or may not associate *acquire* with *acquires*. In this work, we propose a novel *morph-fitting* procedure which moves past the use of curated semantic lexicons for improving distributional vector spaces. Instead, our method injects morphological constraints generated using simple language-specific rules, pulling *inflectional* forms of the same word close together and pushing *derivational antonyms* far apart. In intrinsic evaluation over four languages, we show that our approach: **1)** improves low-frequency word estimates; and **2)** boosts the semantic quality of the entire word vector collection. Finally, we show that morph-fitted vectors yield large gains in the downstream task of *dialogue state tracking*, highlighting the importance of morphology for tackling long-tail phenomena in language understanding tasks.

## 1 Introduction

Word representation learning has become a research area of central importance in natural language processing (NLP), with its usefulness demonstrated across many application areas such as parsing (Chen and Manning, 2014), machine translation (Zou et al., 2013), and many others (Turian et al., 2010; Collobert et al., 2011). Most promi-

nent word representation techniques are grounded in the *distributional hypothesis*, relying on word co-occurrence information in large textual corpora (Curran, 2004; Turney and Pantel, 2010; Mikolov et al., 2013; Mnih and Kavukcuoglu, 2013; Levy and Goldberg, 2014; Schwartz et al., 2015, i.a.).

Morphologically rich languages, in which "substantial grammatical information. . . is expressed at word level" (Tsarfaty et al., 2010), pose specific challenges for NLP. This is not always considered when techniques are evaluated on languages such as English or Chinese, which do not have rich morphology. In the case of distributional vector space models, morphological complexity brings two challenges to the fore:

**1. Estimating Rare Words:** A single lemma can have many different surface realisations. Naively treating each realisation as a separate word leads to sparsity problems and a failure to exploit their shared semantics. On the other hand, lemmatising the entire corpus can obfuscate the differences that exist between different word forms even though they share some aspects of meaning.

**2. Embedded Semantics:** Morphology can encode semantic relations such as antonymy (e.g. *literate* and *illiterate*, *expensive* and *inexpensive*) or synonymy (*north*, *northern*, *northerly*).

In this work, we tackle the two challenges jointly by introducing a *resource-light* vector space fine-tuning procedure termed *morph-fitting*. The proposed method does not require curated knowledge bases or gold lexicons. Instead, it makes use of the observation that morphology implicitly encodes semantic signals pertaining to synonymy (e.g., German word inflections *katalanisch, katalanischem, katalanischer* denote the same semantic concept in different grammatical roles), and antonymy (e.g., *mature* vs. *immature*), capitalising on the proliferation of word forms in morphologically

| en_expensive | de_teure | it_costoso | en_slow | de_langsam | it_lento | en_book | de_buch | it_libro |
|---|---|---|---|---|---|---|---|---|
| costly | teuren | dispendioso | fast | allmählich | lentissimo | books | sachbuch | romanzo |
| costlier | kostspielige | remunerativo | slower | rasch | lenta | memoir | buches | racconto |
| cheaper | aufwändige | redditizio | slower | gemächlich | inesorabile | novel | romandebüt | volumetto |
| prohibitively | kostenintensive | rischioso | slowed | schnell | rapidissimo | storybooks | büchlein | saggio |
| pricey | aufwendige | costosa | slowing | explosionsartig | graduale | blurb | pamphlet | ecclesiaste |
| expensiveness | teures | costosa | slow | langsamer | lenti | booked | bücher | libri |
| costly | teuren | costose | slowing | langsames | lente | rebook | büch | libra |
| costlier | teurem | costosi | slowed | langsame | lenta | booking | büche | librare |
| ruinously | teurer | dispendioso | slowness | langsamem | veloce | rebooked | büches | libre |
| unaffordable | teurerer | dispendiose | slows | langsamen | rapido | books | büchen | librano |

Table 1: The nearest neighbours of three example words (*expensive*, *slow* and *book*) in English, German and Italian before (top) and after (bottom) morph-fitting.

rich languages. Formalised as an instance of the post-processing *semantic specialisation* paradigm (Faruqui et al., 2015; Mrkšić et al., 2016), morph-fitting is steered by a set of linguistic constraints derived from simple language-specific rules which describe (a subset of) morphological processes in a language. The constraints emphasise similarity on one side (e.g., by extracting *morphological* synonyms), and antonymy on the other (by extracting *morphological* antonyms), see Fig. 1 and Tab. 2.

The key idea of the fine-tuning process is to pull synonymous examples described by the constraints closer together in a transformed vector space, while at the same time pushing antonymous examples away from each other. The explicit post-hoc injection of morphological constraints enables: **a)** estimating more accurate vectors for low-frequency words if they are described by the constraints containing their relation with high-frequency words,[1] thus tackling the data sparsity problem; and **b)** specialising the distributional space to distinguish between similarity and association, thus supporting language understanding applications such as *dialogue state tracking* (DST).

As a post-processor, morph-fitting allows the integration of morphological rules with any distributional vector space in any language: it treats an input distributional word vector space as a black box and fine-tunes it so that the transformed space reflects the knowledge coded in the input morphological constraints (e.g., Italian words *rispettoso* and *irrispetosa* should be far apart in the transformed vector space, see Fig. 1). Tab. 1 illustrates the effects of morph-fitting by qualitative examples in three languages: the vast majority of nearest neighbours are "morphological" synonyms.

We demonstrate the efficacy of morph-fitting in four languages (English, German, Italian, Rus-

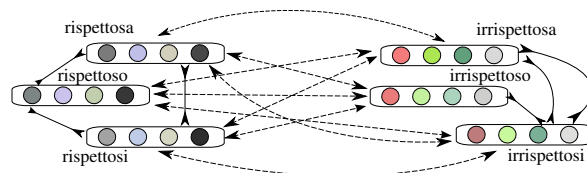

Figure 1: *Morph-fitting* in Italian. Representations for *rispettoso*, *rispettosa*, *rispettosi* (EN: *respectful*), are pulled closer together in the vector space (solid lines; ATTRACT constraints). At the same time, the model pushes them away from their antonyms (dashed lines; REPEL constraints) *irrispettoso*, *irrispettosa*, *irrispettosi* (EN: *disrespectful*), obtained through morphological affix transformation.

sian), yielding large and consistent improvements on benchmarking word similarity evaluation sets such as SimLex-999 (Hill et al., 2015), its multilingual extension (Leviant and Reichart, 2015), and SimVerb-3500 (Gerz et al., 2016). The improvements are reported for all four languages, and with a variety of input distributional spaces, verifying the robustness of the approach.

We then show that incorporating morph-fitted vectors into a state-of-the-art neural-network DST model results in improved tracking performance, especially for morphologically rich languages. We report an improvement of 4% on Italian, and 6% on German when using *morph-fitted* vectors instead of the distributional ones, setting a new state-of-the-art DST performance for the two datasets.[2]

## 2 Morph-fitting: Methodology

**Preliminaries** In this work, we focus on four languages with varying levels of morphological complexity: English (EN), German (DE), Italian (IT), and Russian (RU). These correspond to languages in the Multilingual SimLex-999 dataset. Vocabularies $W_{en}$, $W_{de}$, $W_{it}$, $W_{ru}$ are compiled by retaining all word forms from the four Wikipedias with

---

[1]For instance, the vector for the word *katalanischem* which occurs only 9 times in the German Wikipedia will be pulled closer to the more reliable vectors for *katalanisch* and *katalanischer*, with frequencies of 2097 and 1383 respectively.

[2]There are no readily available DST datasets for Russian.

word frequency over 10, see Tab. 3. We then query these (large) vocabularies using a set of simple language-specific *if-then-else* rules to extract sets of linguistic constraints, see Tab. 2.[3] These constraints (Sect. 2.2) are used as input for the vector space post-processing ATTRACT-REPEL algorithm (outlined in Sect. 2.1).

## 2.1 The ATTRACT-REPEL Model

The ATTRACT-REPEL model is an extension of PARAGRAM, proposed by Wieting et al. (2015). It provides a generic framework for incorporating *similarity* (e.g. *successful* and *accomplished*) and *antonymy* constraints (e.g. *nimble* and *clumsy*) into pre-trained word vectors. Given the initial vector space and collections of ATTRACT and REPEL constraints $A$ and $R$, the model gradually modifies the space to bring the designated word vectors closer together or further apart. The method's cost function consists of three terms. The first term pulls the ATTRACT examples $(x_l, x_r) \in A$ closer together. If $B_A$ denotes the current mini-batch of ATTRACT examples, this term can be expressed as:

$$A(\mathcal{B}_A) = \sum_{(x_l, x_r) \in \mathcal{B}_A} (ReLU(\delta_{att} + \mathbf{x}_l \mathbf{t}_l - \mathbf{x}_l \mathbf{x}_r)$$
$$+ \quad ReLU(\delta_{att} + \mathbf{x}_r \mathbf{t}_r - \mathbf{x}_l \mathbf{x}_r))$$

where $\delta_{att}$ is the similarity margin which determines how much closer synonymous vectors should be to each other than to each of their respective negative examples. $ReLU(x) = \max(0, x)$ is the standard rectified linear unit (Nair and Hinton, 2010). The 'negative' example $\mathbf{t}_i$ for each word $x_i$ in any ATTRACT pair is the word vector *closest* to $\mathbf{x}_i$ among the examples in the current mini-batch (distinct from its target synonym and $\mathbf{x}_i$ itself). This means that this term forces synonymous words from the in-batch ATTRACT constraints to be closer to one another than to any other word in the current mini-batch.

The second term pushes antonyms away from each other. If $(x_l, x_r) \in B_R$ is the current mini-batch of REPEL constraints, this term is:

$$R(\mathcal{B}_R) = \sum_{(x_l, x_r) \in \mathcal{B}_R} (ReLU(\delta_{rpl} + \mathbf{x}_l \mathbf{x}_r - \mathbf{x}_l \mathbf{t}_r)$$
$$+ \quad ReLU(\delta_{rpl} + \mathbf{x}_l \mathbf{x}_r - \mathbf{x}_r \mathbf{t}_r))$$

---

[3] A native speaker is able to easily come up with these sets of morphological rules (or at least with a reasonable subset of rules) without any linguistic training. What is more, the rules for DE, IT, and RU were created by non-native, non-fluent speakers with a limited knowledge of the three languages, exemplifying the simplicity and portability of the approach.

| English | German | Italian |
|---|---|---|
| (discuss, discussed) | (schottisch, schottischem) | (golfo, golfi) |
| (laugh, laughing) | (damalige, damaligen) | (minato, minata) |
| (pacifist, pacifists) | (kombiniere, kombinierte) | (mettere, metto) |
| (evacuate, evacuated) | (schweigt, schweigst) | (crescono, cresci) |
| (evaluate, evaluates) | (hacken, gehackt) | (crediti, credite) |
| (dressed, undressed) | (stabil, unstabil) | (abitata, inabitato) |
| (similar, dissimilar) | (geformtes, ungeformt) | (realtà, irrealtà) |
| (formality, informality) | (relevant, irrelevant) | (attuato, inattuato) |

Table 2: Example synonymous (inflectional; top) and antonymous (derivational; bottom) constraints.

| | $|W|$ | $|A|$ | $|R|$ |
|---|---|---|---|
| English | 1,368,891 | 231,448 | 45,964 |
| German | 1,216,161 | 648,344 | 54,644 |
| Italian | 541,779 | 278,974 | 21,400 |
| Russian | 950,783 | 408,400 | 32,174 |

Table 3: Vocabulary sizes and counts of ATTRACT ($A$) and REPEL ($R$) constraints.

In this case, each word's 'negative' example is the (in-batch) word vector furthest away from it (and distinct from the word's target antonym). The intuition is that we want antonymous words from the input REPEL constraints to be *further away* from each other than from any other word in the current mini-batch; $\delta_{rpl}$ is now the *repel* margin.

The final term of the cost function serves to retain the abundance of semantic information encoded in the starting distributional space. If $\mathbf{x}_i^{init}$ is the initial distributional vector and $V(\mathcal{B})$ is the set of all vectors present in the given mini-batch, this term (per mini-batch) is expressed as:

$$R(\mathcal{B}_A, \mathcal{B}_R) = \sum_{\mathbf{x}_i \in V(\mathcal{B}_A \cup \mathcal{B}_R)} \lambda_{reg} \left\| \mathbf{x}_i^{init} - \mathbf{x}_i \right\|_2$$

where $\lambda_{reg}$ is the L2 regularisation constant.[4] This term effectively *pulls* word vectors towards their initial (distributional) values, ensuring that relations encoded in initial vectors persist as long as they do not contradict the newly injected ones.

## 2.2 Language-Specific Rules and Constraints

**Semantic Specialisation with Constraints** The fine-tuning ATTRACT-REPEL procedure is entirely driven by the input ATTRACT and REPEL sets of constraints. These can be extracted from a variety of semantic databases such as WordNet (Fellbaum, 1998), the Paraphrase Database (Ganitkevitch et al., 2013; Pavlick et al., 2015), or BabelNet (Navigli and Ponzetto, 2012; Ehrmann et al., 2014) as done in prior work (Faruqui et al., 2015; Wieting et al.,

---

[4] We use hyperparameter values $\delta_{att} = 0.6$, $\delta_{rpl} = 0.0$, $\lambda_{reg} = 10^{-9}$ from prior work without fine-tuning. We train all models for 10 epochs with AdaGrad (Duchi et al., 2011).

2015; Mrkšić et al., 2016, i.a.). In this work, we investigate another option: extracting constraints *without* curated knowledge bases in a spectrum of languages by exploiting inherent language-specific properties related to linguistic morphology. This relaxation ensures a wider portability of ATTRACT-REPEL to languages and domains without readily available or adequate resources.

**Extracting ATTRACT Pairs** For the ATTRACT constraints, we focus on *inflectional* rather than on *derivational morphology* rules as the former preserve the full meaning of a word, modifying it only to reflect grammatical roles (e.g., verb tense, case markers; *(read, reads)*).[5] This choice is guided by our intent to fine-tune the original vector space to improve the embedded semantic relations.

We define two rules for English, widely recognised as morphologically simple (Avramidis and Koehn, 2008; Cotterell et al., 2016). These are: **(R1)** *if* $w_1, w_2 \in W_{en}$, where $w_2 = w_1 + $ *ing/ed/s*, *then* add $(w_1, w_2)$ and $(w_2, w_1)$ to the set of ATTRACT constraints $A$. This rule yields pairs such as *(look, looks), (look, looking), (look, looked)*.

If $w[:-1]$ is a function which strips the last character from word $w$, the second rule is: **(R2)** *if* $w_1$ ends with the letter *e and* $w_1 \in W_{en}$ *and* $w_2 \in W_{en}$, where $w_2 = w_1[:-1] + $ *ing/ed/s*, *then* add $(w_1, w_2)$ and $(w_2, w_1)$ to $A$. This creates pairs such as *(create, creates), (create, creating)* and *(create, created)*. Naturally, introducing more sophisticated rules is possible in order to cover for other special cases and morphological irregularities (e.g., *sweep / swept*), but in all our EN experiments, $A$ is based on the two simple EN rules R1 and R2.

The other three languages, with more complicated morphology, yield a larger number of rules. In Italian, we rely on the sets of rules spanning: (1) regular formation of plural *(libro / libri)*; (2) regular verb conjugation *(aspettare / aspettiamo)*; (3) regular formation of past participle *(aspettare / aspettato)*; and (4) rules regarding grammatical gender *(bianco / bianca)*. Besides these, another set of rules is used for German and Russian: (5) regular declension (e.g., *asiatisch / asiatischem*).

---

[5]The core difference between *inflectional* and *derivational morphology* may be summarised in a few lines as follows: the former refers to a set of processes through which the word form expresses meaningful syntactic information, e.g., verb tense, without any change to the semantics of the word. On the other hand, the latter refers to the formation of new words with semantic shifts in meaning (Schone and Jurafsky, 2001; Haspelmath and Sims, 2013; Lazaridou et al., 2013; Zeller et al., 2013; Cotterell and Schütze, 2017).

**Extracting REPEL Pairs** As another source of implicit semantic signals, $W$ also contains words which represent *derivational antonyms*: e.g., two words that denote concepts with opposite meanings, generated through a derivational process. We use a standard set of EN "antonymy" prefixes: $AP_{en} = $ *{dis, il, un, in, im, ir, mis, non, anti}* (Fromkin et al., 2013). If $w_1, w_2 \in W_{en}$, where $w_2$ is generated by adding a prefix from $AP_{en}$ to $w_1$, *then* $(w_1, w_2)$ and $(w_2, w_1)$ are added to the set of REPEL constraints $R$. This rule generates pairs such as *(advantage, disadvantage)* and *(regular, irregular)*. An additional rule replaces the suffix *-ful* with *-less*, extracting antonyms such as *(careful, careless)*.

Following the same principle, we use $AP_{de} = $ *{un, nicht, anti, ir, in, miss}*, $AP_{it} = $ *{in, ir, im, anti}*, and $AP_{ru} = $ *{не, анти}*. For instance, this generates an IT pair *(rispettoso, irrispettoso)* (see Fig. 1). For DE, we use another rule targeting suffix replacement: *-voll* is replaced by *-los*.

We further expand the set of REPEL constraints by transitively combining antonymy pairs from the previous step with inflectional ATTRACT pairs. This step yields additional constraints such as *(rispettosa, irrispettosi)* (see Fig. 1). The final $A$ and $R$ constraint counts are given in Tab. 3. The full sets of rules are available as supplemental material.

## 3 Experimental Setup

**Training Data and Setup** For each of the four languages we train the skip-gram with negative sampling (SGNS) model (Mikolov et al., 2013) on the latest Wikipedia dump of each language. We induce 300-dimensional word vectors, with the frequency cut-off set to 10. The vocabulary sizes $|W|$ for each language are provided in Tab. 3.[6] We label these collections of vectors SGNS-LARGE.

**Other Starting Distributional Vectors** We also analyse the impact of *morph-fitting* on other collections of well-known EN word vectors. These vectors have varying vocabulary coverage and are trained with different architectures. We test standard distributional models: Common-Crawl GloVe (Pennington et al., 2014), SGNS vectors (Mikolov et al., 2013) with various contexts (*BOW* = bag-of-words; *DEPS* = dependency contexts), and training data (*PW* = Polyglot Wikipedia from Al-Rfou

---

[6]Other SGNS parameters were set to standard values (Baroni et al., 2014; Vulić and Korhonen, 2016b): 15 epochs, 15 negative samples, global learning rate: .025, subsampling rate: $1e - 4$. Similar trends in results persist with $d = 100, 500$.

et al. (2013); $8B$ = 8 billion token `word2vec` corpus), following (Levy and Goldberg, 2014) and (Schwartz et al., 2015). We also test the symmetric-pattern based vectors of Schwartz et al. (2016) (*SymPat-Emb*), count-based PMI-weighted vectors reduced by SVD (Baroni et al., 2014) (*Count-SVD*), a model which replaces the context modelling function from CBOW with bidirectional LSTMs (Melamud et al., 2016) (*Context2Vec*), and two sets of EN vectors trained by injecting multilingual information: *BiSkip* (Luong et al., 2015) and *MultiCCA* (Faruqui and Dyer, 2014). We also experiment with a selection of standard distributional spaces in other languages from prior work (Dinu et al., 2015; Luong et al., 2015; Vulić and Korhonen, 2016a).

**Morph-fixed Vectors** A baseline which utilises an equal amount of knowledge as morph-fitting, termed *morph-fixing*, fixes the vector of each word to the distributional vector of its most frequent inflectional synonym, tying the vectors of low-frequency words to their more frequent inflections. For each word $w_1$, we construct a set of $M + 1$ words $W_{w_1} = \{w_1, w'_1, \ldots, w'_M\}$ consisting of the word $w_1$ itself and all $M$ words which co-occur with $w_1$ in the ATTRACT constraints. We then choose the word $w'_{max}$ from the set $W_{w_1}$ with the maximum frequency in the training data, and fix all other word vectors in $W_{w_1}$ to its word vector. The morph-fixed vectors (MFIX) serve as our primary baseline, as they outperformed another straightforward baseline based on *stemming* across all of our intrinsic and extrinsic experiments.

**Morph-fitting Variants** We analyse two variants of morph-fitting: (1) using ATTRACT constraints only (MFIT-A), and (2) using both ATTRACT and REPEL constraints (MFIT-AR).[7]

## 4 Intrinsic Evaluation: Word Similarity

**Evaluation Setup and Datasets** The first set of experiments intrinsically evaluates *morph-fitted* vector spaces on word similarity benchmarks, using Spearman's rank correlation as the evaluation metric. First, we use the SimLex-999 dataset, as well as SimVerb-3500, a recent EN verb pair similarity dataset providing similarity ratings for 3,500 verb

pairs.[8] SimLex-999 was translated to DE, IT, and RU by Leviant and Reichart (2015), and they crowd-sourced similarity scores from native speakers. We use this dataset for our multilingual evaluation.[9]

**Morph-SimLex** We also introduce a synthetic dataset based on multilingual SimLex, termed *Morph-SimLex*. Since the original sets contain only word lemmas, they are unable to evaluate whether a representation model improves vectors for all synonymous word inflections. Therefore, we enrich the sets of pairs using the same set of ATTRACT rules from Sect. 2.2. In short, given a word pair $(w_1, w_2)$ with a SimLex score $sl_{1,2}$, we again construct sets $W_{w_1} = \{w_1, w'_1, \ldots, w'_M\}$ and $W_{w_2} = \{w_2, w''_1, \ldots, w''_N\}$, where $W_{w_1}$ consists of $w_1$ and all words which co-occur with $w_1$ in the $A$ constraints; the same holds for $W_{w_2}$. Morph-SimLex pairs are then generated by taking the Cartesian product between $W_{w_1}$ and $W_{w_2}$, and assigning the same score $sl_{1,2}$ to each such pair. The final dataset is constructed by repeating the procedure for each of the 999 SimLex pairs, yielding 13,213 EN pairs, 17,021 DE pairs, 18,281 IT pairs, and 10,289 RU pairs. We make this dataset available in hope it can aid further research on improving morphological relations in vector spaces.

**Morph-fitting EN Word Vectors** As the first experiment, we morph-fit a wide spectrum of EN distributional vectors induced by various architectures (see Sect. 3). The results on SimLex and SimVerb are summarised in Tab. 4. The results with EN SGNS-LARGE vectors are shown in Fig. 2a. Morph-fitted vectors bring consistent improvement across all experiments, regardless of the quality of the initial distributional space. This finding confirms that the method is robust: its effectiveness does not depend on the architecture used to construct the initial space. To illustrate the improvements, note that the best score on SimVerb for a model trained on running text is achieved by *Context2vec* ($\rho = 0.388$); injecting morphological constraints into this vector space results in a gain of 7.1 $\rho$ points.

**Experiments on Other Languages** We next extend our experiments to other languages, testing both morph-fitting variants. The results are sum-

---

[7]We also tried using another post-processing model (Mrkšić et al., 2016) in lieu of ATTRACT-REPEL. However, this model was computationally intractable with SGNS-LARGE vectors. Moreover, it was consistently outperformed by ATTRACT-REPEL on vector spaces with smaller vocabularies.

[8]Unlike other gold standard resources such as WordSim-353 (Finkelstein et al., 2002) or MEN (Bruni et al., 2014), SimLex and SimVerb provided explicit guidelines to discern between semantic similarity and association, so that related but non-similar words (e.g. *cup* and *coffee*) have a low rating.

[9]Since Leviant and Reichart (2015) re-scored the original EN SimLex, we use their EN SimLex version for consistency.

| | Evaluation | |
|---|---|---|
| Vectors | SimLex-999 | SimVerb-3500 |
| 1. SG-BOW2-PW (300) (Mikolov et al., 2013) | $.339 \rightarrow$ **.439** | $.277 \rightarrow$ **.381** |
| 2. GloVe-6B (300) (Pennington et al., 2014) | $.324 \rightarrow$ **.438** | $.286 \rightarrow$ **.405** |
| 3. Count-SVD (500) (Baroni et al., 2014) | $.267 \rightarrow$ **.360** | $.199 \rightarrow$ **.301** |
| 4. SG-DEPS-PW (300) (Levy and Goldberg, 2014) | $.376 \rightarrow$ **.434** | $.313 \rightarrow$ **.418** |
| 5. SG-DEPS-8B (500) (Bansal et al., 2014) | $.373 \rightarrow$ **.441** | $.356 \rightarrow$ **.473** |
| 6. MultiCCA-EN (512) (Faruqui and Dyer, 2014) | $.314 \rightarrow$ **.391** | $.296 \rightarrow$ **.354** |
| 7. BiSkip-EN (256) (Luong et al., 2015) | $.276 \rightarrow$ **.356** | $.260 \rightarrow$ **.333** |
| 8. SG-BOW2-8B (500) (Schwartz et al., 2015) | $.373 \rightarrow$ **.440** | $.348 \rightarrow$ **.441** |
| 9. SymPat-Emb (500) (Schwartz et al., 2016) | $.381 \rightarrow$ **.442** | $.284 \rightarrow$ **.373** |
| 10. Context2Vec (600) (Melamud et al., 2016) | $.371 \rightarrow$ **.440** | $.388 \rightarrow$ **.459** |

Table 4: The impact of morph-fitting (MFIT-AR used) on a representative set of EN vector space models. All results show the Spearman's $\rho$ correlation before and after morph-fitting. The numbers in parentheses refer to the vector dimensionality.

| Vectors | Distrib. | MFIT-A | MFIT-AR |
|---|---|---|---|
| EN: GloVe-6B (300) | .324 | .376 | **.438** |
| EN: SG-BOW2-PW (300) | .339 | .385 | **.439** |
| DE: SG-DEPS-PW (300) (Vulić and Korhonen, 2016a) | .267 | .318 | **.325** |
| DE: BiSkip-DE (256) (Luong et al., 2015) | .354 | .414 | **.421** |
| IT: SG-DEPS-PW (300) (Vulić and Korhonen, 2016a) | .237 | .351 | **.391** |
| IT: CBOW5-Wacky (300) (Dinu et al., 2015) | .363 | .417 | **.446** |

Table 5: Results on multilingual SimLex-999 (EN, DE, and IT) with two morph-fitting variants.

marised in Tab. 5, while Fig. 2a-2d show results for the morph-fitted SGNS-LARGE vectors. These scores confirm the effectiveness and robustness of morph-fitting across languages, suggesting that the idea of fitting to morphological constraints is indeed language-agnostic, given the set of language-specific rule-based constraints. Fig. 2 also demonstrates that the morph-fitted vector spaces consistently outperform the morph-fixed ones.

Morph-SimLex performance across all languages shows even stronger relative gains over distributional and morph-fixed vectors. The original SimLex dataset only contains word lemmas. Consequently, it fails to penalise word vector collections with bad estimates of less-frequent word forms. The comparison between MFIT-A and MFIT-AR indicates that both sets of constraints are important

for the fine-tuning process: while MFIT-A already yields consistent gains over the initial spaces, a further refinement can be achieved by also incorporating the antonymous REPEL constraints.

# 5 Downstream Task: Dialogue State Tracking (DST)

Goal-oriented dialogue systems provide conversational interfaces for tasks such as booking flights or finding restaurants. In *slot-based* systems, application domains are specified using *ontologies* that define the search constraints which users can express. An ontology consists of a number of *slots* and their assorted *slot values*. In a *restaurant search* domain, sets of slot-values could include PRICE = [*cheap, expensive*] or FOOD = [*Thai, Indian, ...*]. The DST model is the first component of modern dialogue pipelines (Young, 2010). It serves to capture the intents expressed by the user at each dialogue turn and update the *belief state*. This is the system's internal estimate of the user's goals, used by the downstream *dialogue manager* to choose the system response. The following example shows the true dialogue state in a multi-turn dialogue:

> **User:** What's good in the southern part of town?
> `inform(area=south)`
> **System:** Vedanta is the top-rated Indian place.
> **User:** How about something cheaper?
> `inform(area=south, price=cheap)`
> **System:** Seven Days is very popular. Great hot pot.
> **User:** What's the address?
> `inform(area=south, price=cheap); request(address)`
> **System:** Seven Days is at 66 Regent Street.

The Dialogue State Tracking Challenge (DSTC) shared task series formalised the evaluation and provided labelled DST datasets (Henderson et al., 2014a,b; Williams et al., 2016). While a plethora of DST models are available based on, e.g., hand-crafted rules (Wang et al., 2014) or conditional random fields (Lee and Eskenazi, 2013), the recent DST methodology has seen a shift towards neural-network architectures (Henderson et al., 2014c; Mrkšić et al., 2015; Liu and Perez, 2017, i.a.)

**Model: Neural Belief Tracker** To detect intents in user utterances, most existing models rely on either (or both): **1)** Spoken Language Understanding models which require large amounts of annotated training data; or **2)** hand-crafted, domain-specific lexicons which try to capture lexical and morphological variation. The Neural Belief Tracker (NBT) is a novel DST model which overcomes both issues

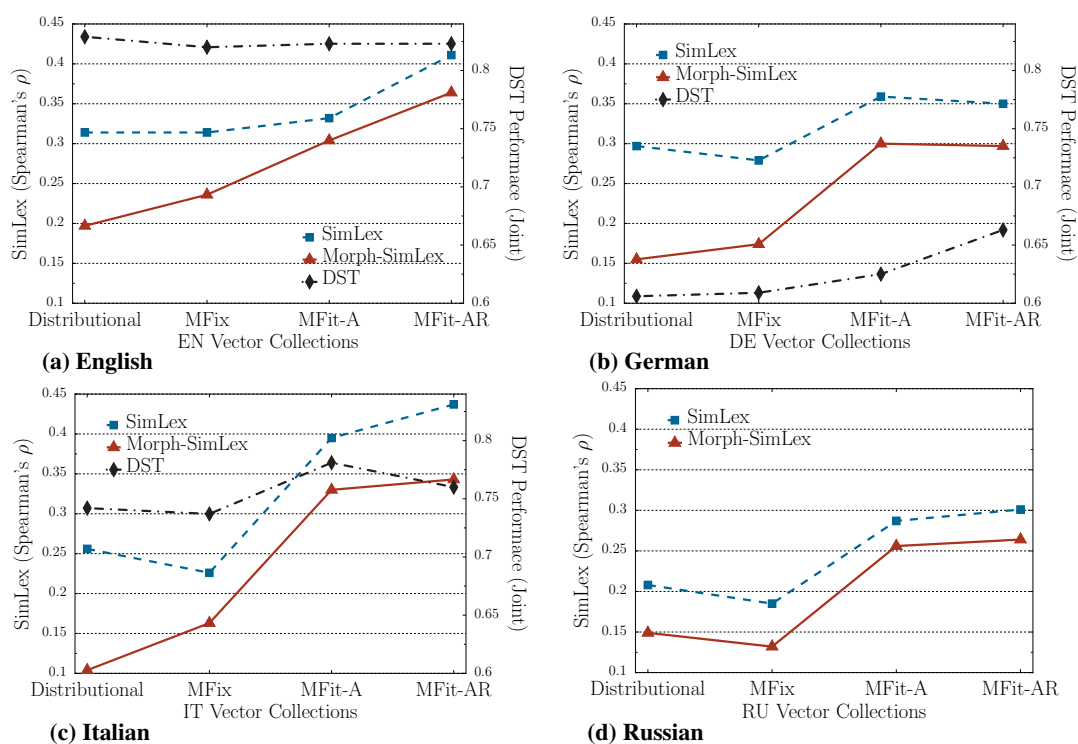

Figure 2: An overview of the results (Spearman's $\rho$ correlation) for four languages on SimLex-999 (blue squares), Morph-SimLex-999 (red triangles), and the downstream DST performance (black diamonds) using SGNS-LARGE vectors ($d = 300$), see Tab. 3 and Sect. 3. The left $y$ axis measures the intrinsic word similarity performance, while the right $y$ axis provides the scale for the DST performance.

by reasoning purely over pre-trained word vectors (Mrkšić et al., 2016). The NBT learns to compose these vectors into intermediate utterance and context representations. These are then used to decide which of the ontology-defined intents (goals) have been expressed by the user. The NBT model keeps word vectors *fixed* during training, so that unseen, yet related words can be mapped to the right intent at test time (e.g. *northern* to *north*).

**Data: Multilingual WOZ 2.0 Dataset** Our DST evaluation is based on the WOZ dataset, released by Wen et al. (2017). In this Wizard-of-Oz setup, two Amazon Mechanical Turk workers assumed the role of the user and the system asking/providing restaurant information. Users typed instead of speaking, removing the need to deal with noisy speech recognition. In DSTC datasets, users would quickly adapt to the system's inability to deal with complex queries. Conversely, the WOZ setup allowed them to use sophisticated language. The WOZ 2.0 release expanded the dataset to 1,200 dialogues (Mrkšić et al., 2016). In this work, we use translations of this dataset to Italian and German, provided by the authors of the original dataset.

**Evaluation Setup** The principal metric we use to

measure DST performance is *joint goal accuracy*, which represents the proportion of test set dialogue turns where all user goals expressed up to that point of the dialogue were decoded correctly (Henderson et al., 2014a). The NBT models for EN, DE and IT are trained using four variants of the SGNS-LARGE vectors: **1)** the initial distributional vectors; **2)** *morph-fixed*; **3)** and **4)** the two variants of *morph-fitted* vectors (see Sect. 3).

**Results and Discussion** The diamond-dashed lines (against the right axes) in Fig. 2 show the DST performance of NBT models making use of the four vector collections. IT and DE benefit from both kinds of *morph-fitting*: IT performance increases $74.1 \rightarrow 78.1$ (MFIT-A) and DE performance rises even more: $60.6 \rightarrow 66.3$ (MFIT-AR), setting a new state-of-the-art score for both languages. The *morph-fixed* vectors do not enhance DST performance, probably because fixing word vectors to their highest frequency inflectional form eliminates useful semantic content encoded in the original vectors. On the other hand, morph-fitting makes use of this information, supplementing it with semantic relations between different morphological forms. These conclusions are in line with the SimLex and Morph-SimLex gains, where morph-fitting outper-

forms distributional and *morph-fixed* vectors.

English performance shows little variation across the four word vector collections investigated here. This corroborates our intuition that, as a morphologically simpler language, English stands to gain less from fine-tuning the morphological variation for downstream applications. This result again points at the discrepancy between intrinsic and extrinsic evaluation: the considerable gains in SimLex performance do not necessarily induce similar gains in downstream performance.

## 6 Related Work

**Semantic Specialisation** A standard approach to incorporating external information into vector spaces is to pull the representations of similar words closer together. Some models integrate such constraints into the training procedure, modifying the prior or the regularisation (Yu and Dredze, 2014; Xu et al., 2014; Bian et al., 2014; Kiela et al., 2015), or using a variant of the SGNS-style objective (Liu et al., 2015; Osborne et al., 2016). Another class of models, popularly termed *retrofitting*, injects lexical knowledge from available semantic databases (e.g., WordNet, PPDB) into pre-trained word vectors (Faruqui et al., 2015; Jauhar et al., 2015; Wieting et al., 2015; Nguyen et al., 2016; Mrkšić et al., 2016). Morph-fitting falls into the latter category. However, instead of resorting to curated knowledge bases, and experimenting solely with English, we show that the *morphological richness* of any language can be exploited as a source of inexpensive supervision for fine-tuning vector spaces, at the same time specialising them to better reflect true semantic similarity.

**Word Vectors and Morphology** The use of morphological resources to improve the representations of morphemes and words is an active area of research. The majority of proposed architectures encode morphological information, provided either as gold standard morphological resources (Sylak-Glassman et al., 2015) such as CELEX (Baayen et al., 1995) or as an external analyser such as Morfessor (Creutz and Lagus, 2007), along with distributional information jointly at *training* time in the language modelling (LM) objective (Luong et al., 2013; Botha and Blunsom, 2014; Qiu et al., 2014; Cotterell and Schütze, 2015; Bhatia et al., 2016, i.a.). The key idea is to learn a morphological composition function (Lazaridou et al., 2013; Cotterell and Schütze, 2017) which synthesises the

representation of a word given the representations of its constituent morphemes. Contrary to our work, these models typically coalesce all lexical relations.

Another class of models, operating at the character level, shares a similar methodology: such models compose token-level representations from sub-component embeddings (subwords, morphemes, or characters) (dos Santos and Zadrozny, 2014; Ling et al., 2015; Cao and Rei, 2016; Kim et al., 2016; Wieting et al., 2016; Verwimp et al., 2017, i.a.).

In contrast to prior work, our model *decouples* the use of morphological information, now provided in the form of inflectional and derivational rules transformed into linguistic constraints, from the actual training. This pipelined approach results in a simpler, more portable model. In spirit, our work is similar to Cotterell et al. (2016), who formulate the idea of post-training specialisation in a generative Bayesian framework. Their work uses gold morphological lexicons; we show that competitive performance can be achieved using a non-exhaustive set of simple rules. Our framework facilitates the inclusion of *antonyms* at no extra cost and naturally extends to constraints from other sources (e.g., WordNet) in future work. Another practical difference is that we focus on similarity and evaluate morph-fitting in a well-defined downstream task where the artefacts of the distributional hypothesis are known to prompt statistical system failures.

## 7 Conclusion and Future Work

We have presented a novel *morph-fitting* method which injects morphological knowledge in the form of linguistic constraints into word vector spaces. The method makes use of implicit semantic signals encoded in inflectional and derivational rules which describe the morphological processes in a language. The results in intrinsic word similarity tasks show that *morph-fitting* improves vector spaces induced by distributional models across four languages. Finally, we have shown that the use of *morph-fitted* vectors boosts the performance of downstream language understanding models which rely on word representations as features, especially for morphologically rich languages such as German.

Future work will focus on other potential sources of morphological knowledge (Soricut and Och, 2015), porting the framework to other morphologically rich languages and downstream tasks, and on further refinements of the post-processing algorithm and constraints selection.

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
