# Peer review of "Morph-fitting: Fine-Tuning Word Vector Spaces with Simple Language-Specific Rules"

_ACL 2017 — decision unknown_

[Official Review · Reviewer 1 · rating 4 · confidence 4]
soundness 5 · originality 5 · clarity 5 · impact 3 · substance 4 · appropriateness 5 · meaningful comparison 3 · presentation format Poster

- Strengths:

- nice, clear application of linguistics ideas to distributional semantics
- demonstrate very clear improvements on both intrinsic and extrinsic eval

- Weaknesses:

- fairly straightforward extension of existing retrofitting work
- would be nice to see some additional baselines (e.g. character embeddings)

- General Discussion:

The paper describes "morph-fitting", a type of retrofitting for vector spaces
that focuses specifically on incorporating morphological constraints into the
vector space. The framework is based on the idea of "attract" and "repel"
constraints, where attract constraints are used to pull morphological
variations close together (e.g. look/looking) and repel constraints are used to
push derivational antonyms apart (e.g. responsible/irresponsible). They test
their algorithm on multiple different vector spaces and several language, and
show consistent improvements on intrinsic evaluation (SimLex-999, and
SimVerb-3500). They also test on the extrinsic task of dialogue state tracking,
and again demonstrate measurable improvements over using
morphologically-unaware word embeddings.

I think this is a very nice paper. It is a simple and clean way to incorporate
linguistic knowledge into distributional models of semantics, and the empirical
results are very convincing. I have some questions/comments below, but nothing
that I feel should prevent it from being published.

- Comments for Authors

1) I don't really understand the need for the morph-simlex evaluation set. It
seems a bit suspect to create a dataset using the same algorithm that you
ultimately aim to evaluate. It seems to me a no-brainer that your model will do
well on a dataset that was constructed by making the same assumptions the model
makes. I don't think you need to include this dataset at all, since it is a
potentially erroneous evaluation that can cause confusion, and your results are
convincing enough on the standard datasets.

2) I really liked the morph-fix baseline, thank you for including that. I would
have liked to see a baseline based on character embeddings, since this seems to
be the most fashionable way, currently, to side-step dealing with morphological
variation. You mentioned it in the related work, but it would be better to
actually compare against it empirically.

3) Ideally, we would have a vector space where morphological variants are just
close together, but where we can assign specific semantics to the different
inflections. Do you have any evidence that the geometry of the space you end
with is meaningful. E.g. does "looking" - "look" + "walk" = "walking"? It would
be nice to have some analysis that suggests the morphfitting results in a more
meaningful space, not just better embeddings.

[Official Review · Reviewer 2 · rating 4 · confidence 4]
soundness 5 · originality 5 · clarity 4 · impact 3 · substance 4 · appropriateness 5 · meaningful comparison 3 · presentation format Oral Presentation

The authors propose ‘morph-fitting’, a method that retrofits any given set
of trained word embeddings based on a morphologically-driven objective that (1)
pulls inflectional forms of the same word together (as in ‘slow’ and
‘slowing’) and (2) pushes derivational antonyms apart (as in
‘expensive’ and ‘inexpensive’). With this, the authors aim to improve
the representation of low-frequency inflections of words as well as mitigate
the tendency of corpus-based word embeddings to assign similar representations
to antonyms. The method is based on relatively simple manually-constructed
morphological rules and is demonstrated on both English, German, Italian and
Russian. The experiments include intrinsic word similarity benchmarks, showing
notable performance improvements achieved by applying morph-fitting to several
different corpus-based embeddings. Performance improvement yielding new
state-of-the-art results is also demonstrated for German and Italian on an
extrinsic task - dialog state tracking. 

Strengths:

- The proposed method is simple and shows nice performance improvements across
a number of evaluations and in several languages. Compared to previous
knowledge-based retrofitting approaches (Faruqui et al., 2015), it relies on a
few manually-constructed rules, instead of a large-scale knowledge base, such
as an ontology.

- Like previous retrofitting approaches, this method is easy to apply to
existing sets of embeddings and therefore it seems like the software that the
authors intend to release could be useful to the community.

- The method and experiments are clearly described. 

Weaknesses:

- I was hoping to see some analysis of why the morph-fitted embeddings worked
better in the evaluation, and how well that corresponds with the intuitive
motivation of the authors. 

- The authors introduce a synthetic word similarity evaluation dataset,
Morph-SimLex. They create it by applying their presumably
semantic-meaning-preserving morphological rules to SimLex999 to generate many
more pairs with morphological variability. They do not manually annotate these
new pairs, but rather use the original similarity judgements from SimLex999.
The obvious caveat with this dataset is that the similarity scores are presumed
and therefore less reliable. Furthermore, the fact that this dataset was
generated by the very same rules that are used in this work to morph-fit word
embeddings, means that the results reported on this dataset in this work should
be taken with a grain of salt. The authors should clearly state this in their
paper.

- (Soricut and Och, 2015) is mentioned as a future source for morphological
knowledge, but in fact it is also an alternative approach to the one proposed
in this paper for generating morphologically-aware word representations. The
authors should present it as such and differentiate their work.

- The evaluation does not include strong morphologically-informed embedding
baselines. 

General Discussion:

With the few exceptions noted, I like this work and I think it represents a
nice contribution to the community. The authors presented a simple approach and
showed that it can yield nice improvements using various common embeddings on
several evaluations and four different languages. I’d be happy to see it in
the conference.

Minor comments:

- Line 200: I found this phrasing unclear: “We then query … of linguistic
constraints”.

- Section 2.1: I suggest to elaborate a little more on what the delta is
between the model used in this paper and the one it is based on in Wieting
2015. It seemed to me that this was mostly the addition of the REPEL part.

- Line 217: “The method’s cost function consists of three terms” - I
suggest to spell this out in an equation.

- Line 223:  x and t in this equation (and following ones) are the vector
representations of the words. I suggest to denote that somehow. Also, are the
vectors L2-normalized before this process? Also, when computing ‘nearest
neighbor’ examples do you use cosine or dot-product? Please share these
details.

- Line 297-299: I suggest to move this text to Section 3, and make the note
that you did not fine-tune the params in the main text and not in a footnote.

- Line 327: (create, creates) seems like a wrong example for that rule. 

* I have read the author response